# Synthesis, Photoisomerization, Antioxidant Activity, and Lipid-Lowering Effect of Ferulic Acid and Feruloyl Amides

**DOI:** 10.3390/molecules26010089

**Published:** 2020-12-28

**Authors:** Chiara Lambruschini, Ilaria Demori, Zeinab El Rashed, Leila Rovegno, Elena Canessa, Katia Cortese, Elena Grasselli, Lisa Moni

**Affiliations:** 1Department of Chemistry and Industrial Chemistry, University of Genoa, Via Dodecaneso 31, 16146 Genova, Italy; chiara.lambruschini@unige.it; 2Department of Earth, Environment and Life Science, University of Genoa, Corso Europa 26, 16132 Genova, Italy; ilaria.demori@unige.it (I.D.); zeinab.elrashed@edu.unige.it (Z.E.R.); leila.rovegno@gmail.com (L.R.); 3Rammal Rammal Laboratory (ATAC Group), Faculty of Sciences I, Lebanese University, Beirut 1003, Lebanon; 4MICAMO Spin-Off Department of Earth Sciences, University of Genoa, Corso Europa 26, 16132 Genova, Italy; elenac90@gmail.com; 5DIMES, Department of Experimental Medicine, University of Genoa, Via Antonio de Toni 14, 16132 Genova, Italy; cortesek@unige.it

**Keywords:** natural phenols, photoisomerization, multicomponent reaction, antioxidant, anti-steatotic

## Abstract

The Ugi four-component reaction employing naturally occurred ferulic acid (FA) is proposed as a convenient method to synthesize feruloyl tertiary amides. Applying this strategy, a peptoid-like derivative of ferulic acid (FEF77) containing 2 additional hydroxy-substituted aryl groups, has been synthesized. The influence of the configuration of the double bond of ferulic acid and feruloyl amide on the antioxidant activity has been investigated thanks to light-mediated isomerization studies. At the cellular level, both FA, trans and cis isomers of FEF77 were able to protect human endothelial cord vein (HECV) cells from the oxidative damage induced by exposure to hydrogen peroxide, as measured by cell viability and ROS production assays. Moreover, in steatotic FaO rat hepatoma cells, an in vitro model resembling non-alcoholic fatty liver disease (NAFLD), the molecules exhibited a lipid-lowering effect, which, along with the antioxidant properties, points to consider feruloyl amides for further investigations in a therapeutic perspective.

## 1. Introduction

Ferulic acid (FA) belongs to the phenolic acid group commonly present in plant tissues. It is widely found in fruits, vegetables, whole grains, cereal seeds, coffee, beer [1,2], as well as in numerous non-edible bio-resources such as bagasse, wheat bran, and beetroot pulp, etherified to lignin or esterified to hemicelluloses. FA displays low toxicity, quite efficient intestinal absorption and high stability [3]. These properties, together with the powerful antioxidant activity, render FA a suitable candidate to be considered for the treatment of several oxidative stress-based pathologies such as cancer, cardiomyopathy, skin disorders, brain disorders, viral infections, diabetes, etc. [4]. Some molecular mechanisms underlying FA antioxidant effects have been clarified. FA induces the translocation of NF-E2-related factor (Nrf2) into the nucleus, where it hetero-dimerizes with musculoaponeurotic fibrosarcoma protein (Maf) and binds to antioxidant response element (ARE), thus promoting the transcription of oxidative stress-responsive genes [5]. FA also induces an increase in the expression of superoxide dismutase and catalase in a p38-mediated manner, thus exerting a pro-survival role [6]. Moreover, FA has been proven to modulate proteins belonging to matrix metalloproteinase (MMP) family, which are involved in several disorders [4], and to inhibit mammalian target of rapamycin (mTOR), the master key regulator of autophagy process, controlling several survival or death signaling pathways that may commit the fate of cancer cells [7]. As a member of the family of natural phenols, FA is able to act as an antioxidant intercepting and reacting with free radicals faster than target biomolecules like lipids, fats, and proteins. Several studies have been performed to investigate the mechanism followed by phenolic antioxidants [8]. Generally, two major mechanisms are involved in the antioxidant activity: H-atom abstraction (HAT) and single-electron transfer (SET) followed by deprotonation; in both cases, a relatively stable phenoxyl radical is formed. In principle, all the phenol compounds can be considered antioxidants and act with both mechanisms, but their efficiency is strongly dependent on the substituents present on the aromatic ring. FA contains a methoxy group and a propenoic acid moiety at the ortho and para positions of phenol, respectively. The electron-donating methoxy group might be responsible for the increased antioxidant activity of FA with respect to 4-hydroxycinnamic acid, while the *E* double bond of the propenoic group might play a responsible role in the stabilization of the phenoxyl radical by resonance [9]. Although several natural phenols containing a catechol, or a pyrogallol type rings, present high radical scavenging activity in vitro, their high susceptibility to oxidation makes their half-life in the body strongly reduced and therefore their applicability as active pharmaceutical ingredients limited [10,11]. On the contrary, FA is much more stable to oxidation under physiological conditions and more promising from a pharmacokinetic point of view.

However, FA suffers a relatively low solubility in hydrophobic media, which limits its ability to cross lipid-rich cell membranes and its application as an inhibitor of lipids and fats autoxidation [9]. For this reason, the synthesis of feruloyl derivatives with antioxidant activity and physicochemical tuned properties is highly desirable. Recently, we published a very fast and efficient synthesis of a series of polyphenols containing from 2 to 4 hydroxy-substituted aryl groups, most of them derived from renewable sources such as natural phenols [12,13,14]. Applying our short fragment-based approach, a series of tertiary feruloyl amides were synthesized. We thought that this convergent approach, based on an Ugi multicomponent reaction of phenol-containing simple substrates, represents a convenient tool for the generation of feruloyl derivatives with the aim of modulating the pharmacokinetic properties and biostability and maintaining the antioxidant effect.

Herein, we describe the revisited synthesis of feruloyl amide trans-FEF77 (t-FEF77). FEF77 is a peptoid-like compound containing 2 additional hydroxy-substituted aryl groups, derived from renewable sources, as tyramine and 4-hydroxybenzaldehyde. Among our in-house library of feruloyl amides, we selected FEF77 as a “proof-of-concept compound” to demonstrate if this class of FA derivatives maintains the anti-oxidant and biological properties of FA. Moreover, FEF77 contains an additional monophenolic group and we evaluated its effect. Chemical assays and in vitro cellular models were used to perform a comparative analysis of FA and t-FEF77 antioxidant potentials. In addition, the influence of the configuration of the double bond on the antioxidant activity was investigated thanks to the complete light-mediated isomerization of t-FEF77 into cis-FEF77 (c-FEF77). Finally, the lipid-lowering effect of FA and t-FEF77 was evaluated in a validated in vitro model of hepatic steatosis.

## 2. Results

### 2.1. Synthesis of FEF77

Based on our standard and optimized procedure [12,13] peptoid-like polyphenols can be generated in a three-steps process, involving: (1) Ugi reaction employing phenolic building blocks protected as allyl ethers; (2) removal of the allyl protecting groups and acetylation of the crude; (3) final solvolysis of polyacetates. This general strategy solved the problems that emerged during our preliminary investigation, where, in several cases, free phenolic groups had a negative effect on the yield and cleanliness of the Ugi reaction. In this work, we tried to access our target compounds FEF77 in just one step by employing components containing the free phenols in the Ugi reaction (Scheme 1). Using standard reaction conditions, we could isolate trans-FEF77 after column chromatography in good yield and high HPLC purity.

### 2.2. Study of Stability of Caffeic Acid, FA, and FEF77 in Buffer Solutions

To prove the superior stability of FA and amide-derivative t-FEF77, we monitored them and their catechol analog (caffeic acid) by HPLC. As shown in Figure 1, after 24 h at room temperature in phosphate-buffered saline (PBS), FA is perfectly stable and even after seven days the amount is ~90% with respect to the starting value. On the other hand, caffeic acid (CA) was proven to be highly unstable; actually, after 24 h, a clear decrease of the HPLC area was observed and after one week, the amount of CA still present in solution was only 30%. During the HPLC analysis of t-FEF77, we identified a minor peak, which indicated the presence of a small amount of cis-isomer. Moreover, we found that the amount of cis isomer increased over time if the sample was simply kept on the bench exposed to the natural light of the laboratory, until it became the majority. In any case, the stability of t-FEF77 was completely restored, keeping the sample in the dark (Figure 1).

Although trans–cis photoisomerization of cinnamic acids has been previously documented in the literature [15,16], the process is usually promoted by UV light, which is unlikely to be present inside a laboratory environment. Moreover, FA and CA did not show this behavior when treated in the same conditions. These data led us to deeper investigate the photoisomerization of FEF77.

### 2.3. Light Conversion Study of FEF77

The photoisomerization studies were carried out using three different light sources: 300 nm, 350 nm, and direct sunlight. First, 70 mM solutions of FA and t-FEF77 in CD_3_OD were exposed to the selected sources and NMR spectra were registered over time. The use of NMR allowed us to confirm the structure of the cis isomer for the newly formed molecules and to calculate the trans/cis ratio (Figure 2). Both compounds exhibited the tendency to isomerize, but we observed substantial differences; actually, FEF77 reaches higher conversion than FA when the photostationary state is achieved. At 350 nm, after 60 min, the isomerization of t-FEF77 was complete (94.2% of c-FEF77), while the amount of cis-FA was only 45%. At 300 nm, the conversion of t-FEF77 was lower (83%), whilst that of FA was higher (59%). When the samples were exposed to direct sunlight, we observed a similar behavior to 350 nm, and the complete conversion of FEF77 was achieved after 60 min, while for FA, the amount of cis was only 40% after 120 min.

Having in hand pure c-FEF77, we were able to compare the antioxidant activity of the three compounds. To the best of our knowledge, it is the first time that a complete isomerization of a FA derivative has been observed.

### 2.4. Antioxidant Activity of FA, t-FEF77, and c-FEF77

#### 2.4.1. DPPH Assay and Radical Scavenging Activity

To evaluate the antioxidant activity of FA, t-FEF77, and c-FEF77, we first performed the standard 2,2-diphenyl-1-picrylhydrazyl radical (DPPH) assay. DPPH assay is conducted monitoring the decrease of the absorbance at 515 nm of a solution of DPPH radical in the presence of different concentrations of antioxidant. The molar ratio (MR) and the percentage of radical scavenging activities (RSA) are easily determined as already reported in the literature [17]. Moreover, the half-maximal efficient concentrations (EC_50_) are determined when the system reaches a steady-state (after 240 min) and reported as molar ratio (Table 1).

As expected, the RSA depends on time and on the concentration of the antioxidant. The first experiments were performed monitoring the absorbance at 515 nm of DPPH in the presence of different concentrations of FA and t-FEF77 for 15 min. The results showed in Table 1 revealed that both phenolic compounds presented appreciable DPPH radical scavenging potential, which appeared concentration-dependent. Clearly, the RSA(1) values are slightly lower for t-FEF77. This result is consistent with previous studies, where it has been shown that FA was more effective than the urethane amide derivative [9] or a tyrosine-amide derivative [18] in scavenging DPPH radicals. Since the lower antioxidant efficiency of monophenol rings has been well established in literature [19,20], we decided to carry out the experiments for a longer time, in order to evaluate the contribution of the additional monophenolic groups present in FEF77. Hence, based on RSA(2) values, FA and t-FEF77 showed a similar profile.

The molar EC_50_ was even higher for t-FEF77 than for FA, which shows that the presence of two additional monophenolic rings and a tertiary amide at the end of the propenoic side, do not affect the antioxidant performance of FA. On the other hand, c-FEF77 showed a slightly lower radical scavenging activity, reaching just 47% of RSA(2) at the highest concentration examined.

#### 2.4.2. Antioxidant Activity in HECV Cells

Preliminarily, the effects of FA, t-FEF77, and c-FEF77 on human endothelial cord vein (HECV) cell viability was checked by MTT cell viability assay [21]. Starting from 50 μM, a concentration at which a decrease in cell viability has been reported for FA [22], we halved the doses until reaching 3.125 µM. As shown in Figure 3, none of the three molecules exerted a cytotoxic effect at any of the concentrations tested in the range 3.125–50 μM. Therefore, the intermediate dose of 12.5 µM was selected for further experiments.

Secondly, we tested HECV cell viability after a 2 h treatment with H_2_O_2_ ranging from 0.25 to 1.5 mM. As shown in Figure 4A, at concentrations ≤0.75 mM, H_2_O_2_ significantly reduced cell viability by ≥50% (*p* ≤ 0.001 with respect to control). Therefore, 0.75 mM H_2_O_2_ was the concentration selected for further experiments, in which HECV cells were pretreated with 12.5 μM polyphenols (FA, t-FEF77, and c-FEF77) for 24 h and subsequently with H_2_O_2_ for 2 h. As shown in Figure 4B, polyphenol pretreatment significantly prevented the decrease in cell viability induced by H_2_O_2_ (+25% for both FA and t-FEF77 and +28% for c-FEF77; *p* ≤ 0.001 for all with respect to H_2_O_2_ 0.75 mM, Figure 4B).

The protective properties of FA, t-FEF77, and c-FEF77 against oxidative stress were confirmed by 2′,7′–dichlorofluorescin diacetate (DCF-DA) stain for reactive oxygen species (ROS) detection in HECV cells.

In control cells, FA, t-FEF77, and c-FEF77 treatments did not affect intracellular ROS content (not shown), but prevented the oxidative damage induced by subsequent treatment with H_2_O_2_. These results are demonstrated by the evident decrease in fluorescence depicted in the representative images of Figure 4C.

### 2.5. Lipid-Lowering Effect of FA and t-FEF77

We tested the lipid-lowering effects of FA and FEF77 in a validated in vitro model of hepatic steatosis consisting of FaO rat hepatoma cells incubated for 3 h with oleate/palmitate mixture (O/P; 2:1 molar ratio, final concentration 0.75 mM) [23]. To this aim, we used only t-FEF77, since the trans and cis isomers exhibited a comparable antioxidant ability both in DPPH assay and mostly in the HECV cellular model. In preliminary experiments, we checked the effects of FA and t-FEF77 on FaO cell viability. Neither FA nor t-FEF77 induced any changes in cell viability both in the absence and presence of O/P, as measured by MTT assay (Figure 5A).

The effects of FA and t-FEF77 on triglyceride (TAG) accumulation in O/P FaO cells were then evaluated. As shown in Figure 5B, cell exposure to O/P for 3 h resulted in significant TAG accumulation (+105%; *p* ≤ 0.001 with respect to control), as previously demonstrated [24,25]. FA or t-FEF77 per se did not alter lipid content in control cells (data not shown), but the treatment of O/P cell with FA or t-FEF77 12.5 μM for 24 h significantly reduced intracellular TAG content with respect to O/P (−26%; *p* ≤ 0.01). These results were confirmed by fluorescence microscopy visualization of intracellular lipid droplets stained with BODIPY 493/503. As shown in Figure 5C, only a few green droplets are detectable in control FaO cells, whereas their number increases after O/P treatment. Upon a 24 h incubation with FA or t-FEF77, fewer lipid droplets are detectable as compared to O/P cells.

## 3. Discussion

The trans–cis photoisomerization of cinnamic acids is usually promoted by UV light and generally produces a mixture of isomers. During the light-mediated isomerization, we were able to fully convert t-FEF77 in c-FEF77, which could be isolated in pure form and characterized. Thanks to this unprecedented result, the UV-Vis spectra were registered and compared, showing a general decrease of intensity for c-FEF77. Trans-FEF77 has a maximum at 330 nm that is not present in c-FEF77 and there is an overall blue-shift of the peaks for c-FEF77 in the 250–350 nm region (Appendix A). Our hypothesis for these observations is a partial distortion from planarity for c-FEF77 due to steric hindrance of the tertiary amide and therefore the extended conjugation is no longer present. Based on the UV-Vis spectra, we can also explain the different amounts of c-FEF77 at 350 and 300 nm, since the trans/cis ratio at the photostationary state depends on the relative molar attenuation coefficient (ε) of the two isomers and on the relative quantum yield (Φ) of their photochemical conversion [26]. At 300 nm, both isomers absorb, although t-FEF77 has a higher molar attenuation coefficient and at the photostationary state, a 17:83 ratio is observed. On the other hand, at 350 nm, only t-FEF77 absorbs and therefore a complete isomerization can occur.

Similarly, the distortion from planarity caused by the steric hindrance of the tertiary amide might cause the slightly lower radical scavenging activity of c-FEF77. Actually, the conjugated double bond in the side chain produces a stabilizing effect by resonance on the phenoxyl radical, thus enhancing the antioxidant activity of FA and t-FEF77 with respect to c-FEF77.

As the relative scavenging activity is influenced by analytical conditions [27], other assays have been performed by using biological systems represented by HECV cells [28] in which oxidative damage was induced by incubation with 0.75 mM H_2_O_2_. FA, t-FEF77, and c-FEF77, all significantly prevented H_2_O_2_ injury without differences among these three molecules. Taken together, our results suggest that the different antioxidant activities of FA, t-FEF77, and c-FEF77 as measured by DPPH assay are not appreciable when tested in biological systems, and that all the three compounds exhibit a comparable protective action against oxidative damage in a cellular model.

Oxidative stress and ROS overproduction are considered a basic mechanism underlying the development of different pathologies such as cancer, cardiovascular events, neurodegenerative diseases, and metabolic impairments. Some of us developed a long-time experience in the use of experimental models of non-alcoholic fatty liver disease (NAFLD). NAFLD is a physio-pathological condition affecting 25% of the population worldwide and capable of worsening towards more severe pathologies such as steatohepatitis, fibrosis, cirrhosis, and hepatocellular carcinoma [29]. The pathogenesis of NAFLD is complex, encompassing different mechanisms and multiple coexisting “hits”, among which oxidative stress can be considered as the starting point of the hepatic damage [30]. As a matter of fact, natural anti-oxidant compounds such as vitamin E and plant-derived polyphenols have been investigated and proved to exert beneficial effects in experimental models of NAFLD [31].

FA and methyl FA have been proven to exert hepatoprotective effects against acetaminophen or methotrexate hepatotoxicity, and to reduce hepatic steatosis in alcohol and high fat diet-fed animals [4,32,33]. These observations are in line with our results, demonstrating a comparable lipid-lowering action of both FA and FEF77 in a validated in vitro model of NAFLD. Moreover, we demonstrated a significant protective capacity of FA and FEF77 against the oxidative damage of endothelial cells (HECV). Taken together, these data could be considered beneficial in NAFLD, since sinusoidal endothelium dysfunction is involved in the development of fibrosis and worsening of liver damage [34]. In vitro models of hepatic steatosis are recognized valuable tools to study NAFLD mechanisms directly at the cellular level, keeping the translational value of the observed results, without exposing humans to unnecessary risks. Undoubtedly, information in this field needs to be confirmed by clinical studies. Nevertheless, from a therapeutic perspective, we consider our results of interest, since the development of synthetic products maintaining similar properties to those of the corresponding natural compounds is a difficult but desirable goal, taking into account that no specific pharmacological treatment has been approved for NAFLD so far.

## 4. Materials and Methods

### 4.1. General Remarks

NMR spectra were taken at r.t. in CD_3_OD at 300 MHz (^1^H), and 75 MHz (^13^C), using as an internal standard, the central peak of CD_3_OD (^1^H-NMR in CD_3_OD; 3.310 ppm) or the central peak of CD_3_OD (^13^C in CD_3_OD; 49.00 ppm). Chemical shifts are reported in ppm (δ scale). Peak assignments were made with the aid of gCOSY and gHSQC experiments. In the ABX system, the proton A is considered upfield and B downfield. IR spectra were recorded as solid, oil, or foamy samples, with the ATR (attenuated total reflectance) technique. TLC analyses were carried out on silica gel plates and viewed at UV (λ = 254 nm or 360 nm) and developed with Hanessian stain (dipping into a solution of (NH_4_)_4_MoO_4_·4H_2_O (21 g) and Ce(SO_4_)_2_·4H_2_O (1 g) in H_2_SO_4_ (31 mL) and H_2_O (469 mL) and warming). R_f_ values were measured after an elution of 7–9 cm. Column chromatography was done with the “flash” methodology by using 220–400 mesh silica. Petroleum ether (40–60 °C) is abbreviated PE. All reactions employing dry solvents were carried out under nitrogen. Extractions were always repeated three times and organic extracts were always dried over Na_2_SO_4_ and filtered before evaporation to dryness. Caffeic acid, ferulic acid, and DPPH radical were purchased from Sigma Aldrich Chemical; methanol and acetonitrile (HPLC grade) were hypergrade for LC-MS from Merk or Carlo Erba Reagents.

Photoinduced isomerization of FEF77 were performed with sunlight or a Southern New England Ultraviolet Company Rayonet^®^ apparatus equipped with 8 Iles Optical lamps (300 or 350 nm). The lamps used were cylindrical with a length of 26.5 cm and a power of 8 watts. The sample was fixed at a 10 cm distance from the lamp.

UV-Vis spectra and DPPH assay were performed using Varian Cary^®^ 50 UV-Vis Spectrophotometer.

HPLC analysis was performed on Agilent HP 1100 equipped with a DAD detector (220 nm) and column Phenomenex Gemini 3u C6-Phenyl 110A (4 μm, 3 × 150 mm^2^). Mobile phases were (A) H_2_O + 0.1% TFA and (B) CH_3_CN + 0.1% TFA, gradient from 10% to 100% of B in 15 min, flow 0.34 mL/min, and 26 °C. Mass analysis was performed on Microsaic 4000 MiD^®^ mass spectrometer.

### 4.2. Synthesis of t-FEF77

A solution of 4-hydroxybenzaldehyde (122 mg, 1.00 mmol) in dry methanol (1 mL) was treated with tyramine (138 mg, 1.00 mmol) under nitrogen atmosphere. After 15 min, the solution was treated at room temperature with trans-ferulic acid (194 mg, 1.00 mmol) and methyl isocyanide (64 μL, 1.10 mmol). After 16 h, the mixture was concentrated and the residue was filtered on a short column of silica gel with EtOAc + 3% MeOH to give t-FEF77 (317 mg, 66%), as white foam. R_f_ = 0.29 (EtOAc); ^1^H-NMR (300 MHz, CD_3_OD): δ 7.50 (d, *J* = 15.3 Hz, 1H; CH=), 7.28 (d, *J* = 8.5 Hz, 2H; 2 CH Ar), 7.12 (d, *J* = 1.3 Hz, 1H; CH Ar), 7.03 (dd, *J* = 8.2, 1.3 Hz, 1H; CH Ar), 6.95–6.85 (m, 2H; 2 CH Ar), 6.82 (d, *J* = 8.1 Hz, 1H; CH Ar), 6.75 (d, *J* = 8.4 Hz, 2H; 2 CH Ar), 6.74 (d, *J* = 15.3 Hz, 1H; CH=), 6.66–6.59 (m, 2H; 2 CH Ar), 6.07 (s, 1H; C*H*NH), 3.91 (s, 3H; OCH_3_), 3.70 (ddd, *J* = 15.8, 10.2, 5.9 Hz, 1H; 1 H of CH_2_N), 3.57–3.45 (m, 1H; 1 H of CH_2_N), 2.76 (s, 3H; C*H*_3_NH), 2.54–2.39 (m, 1H; 1 H of CH_2_Ph), 2.17–2.04 ppm (m, 1H; 1 H of CH_2_Ph); ^13^C-NMR (75 MHz, CD_3_OD) = δ 173.5 (C=O), 170.0 (C=O), 159.4 (C quat), 157.1 (C quat), 150.1 (C quat), 149.3 (C quat), 144.6 (CH=), 132.7 (2 CH Ar), 130.7 (2 CH Ar), 128.4 (C quat), 127.2 (C quat), 123.6 (CH Ar), 116.7 (2 CH Ar), 116.5 (CH Ar), 116.3 (2 CH Ar and C quat.), 116.1 (CH=), 111.5 (CH Ar), 63.4 (CHNH), 56.4 (CH_3_O), 48.6 (CH_2_), 37.1 (CH_2_), 26.4 ppm (CH_3_NH); HPLC-MS (ESI+)-UV. Rt: 10.46 min, 477.4 [M + H]^+^; HPLC showed a purity of 96.3%.

### 4.3. Synthesis of c-FEF77

A solution of t-FEF77 (25 mg) in MeOH (3 mL) in a test tube was exposed to direct sun light. After 3 h, the isomerization is complete and the solvent was removed under vacuum. HPLC showed a purity of 94.2%. White foam. R_f_ = 0.29 (EtOAc); ^1^H-NMR (300 MHz, CD_3_OD), mixture of two rotamers (M:m = 57:43): δ 7.30–7.23 (m, 2H; 2 CH Ar (M)), 7.13 (d, *J* = 2.0 Hz, 1H; CH Ar (M)), 7.09 (d, *J* = 1.9 Hz, 1H; CH Ar (m)), 6.96–6.85 (m, 7H; 7 CH Ar (M+m)), 6.84–6.73 (m, 5H; 5 CH Ar (M+m)), 6.73–6.56 (m, 10H; CH Ar and 2 CH= (M+m)), 6.01 (d, *J* = 12.7 Hz, 2H; 2 CH= (M+m)), 5.99 (s, 1H; CHNH (M)), 5.72 (s, 1H; CHNH (m)), 3.85 (s, 3H; O CH_3_ (M)), 3.82 (s, 3H; O CH_3_ (m)), 3.81–3.72 (m, 1H; 1 H of CH_2_N (M)), 3.47–3.37 (m, 2H; CH_2_N (m)), 3.30–3.18 (m, 1H; 1 H of CH_2_N (M)), 2.83–2.69 (m, 1H; 1 H of CH_2_Ph (m)), 2.75 (s, 3H; CH_3_NH (M)), 2.58 (s, 3H; CH_3_NH (m)), 2.43–2.30 (m, 1H; 1 H of CH_2_Ph (M)), 1.85 (ddd, *J* = 13.0, 10.9, 5.7 Hz, 1H; 1 H of CH_2_Ph (M)), 1.65 (td, *J* = 12.1, 4.5 Hz, 1H; 1 H of CH_2_Ph (m)); ^13^C-NMR (75 MHz, CD_3_OD) = δ 173.0 (C=O), 172.6 (C=O), 172.4 (C=O), 172.3 (C=O), 159.4 (C quat.), 159.3 (C quat.), 156.9 (C quat.), 156.7 (C quat.), 149.0 (C quat.), 148.8 (C quat.), 148.8 (C quat.), 148.7 (C quat.), 135.8 (CH=), 135.3 (CH=), 132.7 (2 CH Ar), 132.2 (2 CH Ar), 131.3 (C quat.), 130.6 (C quat.), 130.5 (3 CH Ar), 128.5 (2 CH Ar), 126.8 (C quat.), 126.8 (C quat.), 124.12 (CH Ar), 124.06 (CH Ar), 121.09 (2 CH Ar), 116.7 (2 CH Ar), 116.5 (2 CH Ar), 116.2 (2 C quat.), 116.2 (3 CH Ar), 116.1 (2 CH=), 113.3 (CH Ar), 112.7 (CH Ar), 65.8 (CHNH), 62.8 (CHNH), 56.6 (OCH_3_), 56.4 (OCH_3_), 36.3 (CH_2_), 34.4 (CH_2_), 26.4 (CH_2_), 26.2 (CH_2_); HPLC-MS (ESI+)-UV. Rt: 10.15 min, 477.4 [M + H]^+^; HPLC showed a purity of 94.2%.

### 4.4. Stability Assays

Stock solutions at 25 mM in DMSO of FA, CA, and t-FEF77 were prepared. Work solutions at 0.25 mM in PBS pH 7.4 were made by diluting stock solutions, in such a manner that the final solutions contain 1% of DMSO. HPLC analysis were run at 0, 24, 48, 96 h and 7 days, and meanwhile, the work solutions were kept at room temperature. The stability was evaluated by measuring the integral area of the peak and the integrals were expressed as a percentage of the value at 0 h, which was imposed equal to 100.

### 4.5. Photoisomerization Studies

First, 70 mM solutions of FA and t-FEF77 in CD_3_OD were exposed to different light sources (300, 350 nm, and direct sunlight) and ^1^H-NMR spectra were registered over time until the steady-state was reached (15, 30, 60, 90, and 120 min). The trans/cis ratio was determined by integration.

### 4.6. DPPH Assays

Method A: a 50 μM solution of DPPH (2.90 mL, 145 nmol) in methanol was placed in a cuvette. A solution of FA or t-FEF (with a known concentration from 200 to 800 μM; 100 μL, from 20 to 80 nmol) in methanol was added, and the UV absorbance was monitored versus time at λ = 515 nm with a UV/Vis spectrophotometer. Absorbance values were taken every 0.1 min for 15 min. Experiments were performed at five different concentrations of the antioxidant. Table 1 reports only selected concentrations.

Method B: a 50 μM solution of DPPH (2.90 mL, 145 nmol) in methanol was placed in a cuvette. A solution of FA, t-FEF77, or c-FEF77 (with a known concentration from 200 to 800 μM; 100 μL, from 20 to 80 nmol) in methanol was added, and the UV absorbance was monitored versus time at λ = 515 nm with a UV/Vis spectrophotometer. Absorbance values were taken at 30, 60, 120, 240 min. Experiments were performed at four different concentrations of the antioxidant.

### 4.7. Chemicals

All chemicals, unless otherwise indicated, were supplied by Sigma-Aldrich Corp. (Milan, Italy).

### 4.8. Cell Culture and Treatments

Human endothelial cord vein (HECV) cells (Cell Bank and Culture-GMP-IST-Genoa, Italy) are a human endothelial cell line isolated from the umbilical vein; they were grown in a humidified atmosphere with 5% CO_2_ at 37 °C in Dulbecco’s modified Eagle’s medium high glucose (D-MEM) supplemented with L-glutamine and 10% fetal calf serum (FCS).

FaO cells (European Collection of Authenticated Cell Cultures, Sigma-Aldrich) are a rat hepatoma cell line maintaining hepatocyte-specific markers [35]. Cells were grown in a humidified atmosphere with 5% CO_2_ at 37 °C in Coon’s modified Ham’s F12 medium supplemented with L-glutamine and 10% FCS.

For cell treatments, stock solutions of FA, t-FEF77, and c-FEF77 were made in DMSO at the concentration of 25 mM and store at −20 °C. Stock solutions were serially diluted to working solutions in culture medium. The final concentration of DMSO in working solutions was 0.1% maximum.

Oxidative stress was induced by incubating confluent HECV cells with increasing concentrations of H_2_O_2_ (0.25–1.5 mM) for 2 h. The effects of polyphenols were studied after incubation of sub-confluent HECV cells with FA, t-FEF77, or c-FEF77 at different concentrations (3.125–50 µM) for 24 h.

To induce intracellular lipid accumulation and obtain an in vitro model of hepatic steatosis, sub-confluent FaO cells were treated for 3 h with a mixture of oleate/palmitate (O/P) at a final concentration of 0.75 mM (2:1 molar ratio) [23]. Thereafter, ‘steatotic’ O/P cells were incubated for 24 h with FA or t-FEF77 12.5 μM.

### 4.9. MTT Assay for Determination of Cell Viability

The MTT (3-(4,5-dimethylthiazol-2-yl)-2,5-diphenyltetrazolium bromide) assay is a measure of the effects of treatments on cellular populations. This colorimetric assay is based on the reduction of the yellow tetrazolium MTT into purple formazan crystals by metabolically active and alive cells. MTT was dissolved in PBS at a concentration of 5 mg/mL and filtered through 0.22 μm pores. The working solution was diluted in culture medium at the final concentration of 0.5 mg/mL. The cells were then incubated for 3 h in a humidified atmosphere with 5% CO_2_ at 37 °C. Precipitated formazan was then dissolved in acid-alcohol (0.04 N HCl in 2-propanol) solution and read at 570 nm in a Varian Cary-50Bio spectrophotometer (Agilent, Milan, Italy) [21].

### 4.10. Intracellular ROS Visualization by Fluorescence Microscopy

The oxidation of the cell-permeant 2′-7′ dichlorofluorescin diacetate (DCF-DA, Fluka, Germany) to 2′-7′dichlorofluorescein (DCF) is used for determining in situ the production of H_2_O_2_ and other ROS. A stock solution of DCF-DA (10 mM in DMSO) was prepared and stored at −20 °C in the dark. At the end of the different treatments, cells were loaded with 10 μM DCF-DA in PBS for 30 min at 37 °C in the dark. Then, the cells were washed several times with PBS. Images were acquired in PBS by using an inverted Olympus IX53 microscope (Olympus, Milano, Italy) equipped with a CCD UC30 camera and a digital image acquisition software (cellSens Entry) at 10× magnification.

### 4.11. Intracellular Triglyceride Content

FaO cells were scraped in PBS and lysed. Lipids were extracted using the chloroform/methanol (2:1) method. Triglyceride (TAG) content was measured by using “Triglycerides liquid” kit (Sentinel, Milan, Italy) and quantified spectrophotometrically in a Varian Cary-50Bio spectrophotometer (Agilent, Milan, Italy). Values were normalized to protein content, as measured by Bradford assay [36]. Data are expressed as percent TAG content relative to controls [37].

### 4.12. Lipid Droplet Visualization by Fluorescence Microscopy

Cells grown on coverslips were rinsed with PBS (pH 7.4) and fixed with 4% paraformaldehyde for 20 min at room temperature. Neutral lipids were stained by incubation with 1 μg/mL BODIPY 493/503 (Molecular Probes, Life technologies, Monza, Italy) in PBS for 30 min [38]. After washing, nuclei were stained and slides were mounted with 4′,6-diamidino-2-phenylindole (DAPI, 5 μg/mL) (ProLong Gold medium with DAPI; Invitrogen). Mounted slides were examined by Nikon Eclipse E80i light microscope (Nikon, Tokyo, Japan) equipped with the standard epifluorescence filter set up. Images were captured under oil with a 100× plan apochromatic objective. Analyses were performed on two independent experiments measuring at least 40 cells for each treatment using the ImageJ software [39].

## 5. Conclusions

Oxidative stress rules a plethora of pathological processes, thus research on polyphenolic compounds is of utmost importance in the hope of discovering new therapeutic agents against such threats to human health. Data presented in this study have shown that tertiary feruloyl amides as trans-FEF77 possess similar antioxidant activity and lipid-lowering effect compared to natural ferulic acid. This finding opens the way for studying specific structural modifications of FA obtaining a fine-tuning of physicochemical properties. Moreover, to the best of our knowledge, this paper has reported the first complete light-mediated isomerization of a ferulic derivative.

## Data Availability

Not applicable.

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
