# Peer review of "Synthesis, Photoisomerization, Antioxidant Activity, and Lipid-Lowering Effect of Ferulic Acid and Feruloyl Amides"

_molecules, 2020, doi:10.3390/molecules26010089_

Round 1
Reviewer 1 Report
The manuscript "Synthesis, photoisomerization, antioxidant activity and lipid-lowering effect of ferulic acid and feruloyl amides" by Moni, Grasselli and co-workers describes the synthesis of novel feruloyl amide derivatives with antioxidant activity and lipid-lowering effect. This interesting work is a logical continuation of the previous research of this scientific group in the field of synthesis of new compounds with biological activity.
The manuscript is well written. However, I have questions and comments on the paper.
1) It is necessary to add information about the main advantage of the new FEF77 molecules. When reading the manuscript, I concluded that the properties of the compounds FEF77 are not better than the starting FA.
2) Have the authors checked the possibility of the reverse transition of the cis-form of FEF77 into the trans-form in time and other conditions?
3) Need to correct - line 300"(75 MHz, cd3od)" to " (75 MHz, CD3OD) "; line 292 "methyl isocyanide (64 L, 1.10 mmol)"; the subscripts in the formulas, for example line 271 "(NH4)4MoO4·4H2O (21 g) and Ce(SO4)2·4H2O"
4) In part 4.1. General remarks the authors provide information about the spectra in CDCl3 or in d6-DMSO, but further in parts 4.2 and 4.3 there are spectra only in CD3OD.
5) "At 350 nm after 120 min the isomerization of t-FEF77 was complete (94.2% of c-FEF77)…" But isomerization takes 60 minutes, and then the amount does not change.
6) "To the best of our knowledge, it is the first time that a complete isomerization of a FA 130 derivative has been observed." It is necessary to add an explanation in what is the uniqueness of the obtained structure of FEF77, which allowed complete isomerization?
7) Table 1. Why is RSA(1) = 20.3 greater than RSA(2) = 16.7 for the FA concentration of 200 M (need to be corrected by μM?)?
8) Need to add the correspondence of images to the letters a), b), c) in the caption of Figure 4. Scale bars of images on Figure 4c are not visible. In Figure 4c, twice labeled 0.75 mM H2O2 + FA 12.5 µM.
9) Supplementary materials should be added the information on the conditions for recording UV and NMR spectra (concentrations and solvents).
10) Contributions of all authors should be added.
Author Response
Genova, December 23rd 2020
Dear Editor,
please find enclosed the revised version of the manuscript “Synthesis, photoisomerization, antioxidant activity and lipid-lowering effect of ferulic acid and feruloyl amides” emended according to reviewers’ suggestions. We thank the reviewers for their constructive criticism. A point by point answer to their comments follows.
Referee 1
- It is necessary to add information about the main advantage of the new FEF77 molecules. When reading the manuscript, I concluded that the properties of the compounds FEF77 are not better than the starting FA.
Answer: As we explained in lines 62-71, although FA is stable respect to other natural phenols, it suffers a relatively low solubility in hydrophobic media, which hamper its application. Therefore the synthesis of feruloyl derivatives may overcome this limitation. We had an in-house library of feruoyl tertiary amides synthetized by Ugi reaction and we decided to investigate if this type of structure modifications do not alter the anti-oxidant and biological properties of FA, with the future aim to tune the pharmacokinetic properties with an optimal choice of substituents. Among all of our feruloyl amides we selected FEF77 as “proof-of-concept compound” and also because we wanted to investigate the effect of an additional monophenol group. Since this part of our goal is not clear, as highlighted by the Referee, we added this sentence (highlighted in yellow in the text) starting from line 78 of the revised manuscript:
“Among our in-house library of feruloyl amides, we selected FEF77 as “proof-of-concept compound” to demonstrate if this class of FA derivatives maintains the anti-oxidant and biological properties of FA. Moreover FEF77 contains an additional monophenolic group and we evaluated its effect.”
- Have the authors checked the possibility of the reverse transition of the cis-form of FEF77 into the trans-form in time and other conditions?
Answer: Yes, we had performed this type of experiments on a model compound and the data will be disclosed in a paper we are preparing. We have proved that cis-feruloyl amides are configurational stable over time and in the following conditions: i) DMSO at 150 °C, ii) Pd/C at rt, iii) strong basic conditions and iv) strong acidic conditions.
- Need to correct - line 300"(75 MHz, cd3od)" to " (75 MHz, CD3OD) "; line 292 "methyl isocyanide (64 L, 10 mmol)"; the subscripts in the formulas, for example line 271 "(NH4)4MoO4·4H2O (21 g) and Ce(SO4)2·4H2O"
Answer: Done, highlighted in yellow
- In part 1. General remarks the authors provide information about the spectra in CDCl3 or in d6-DMSO, but further in parts 4.2 and 4.3 there are spectra only in CD3OD.
Answer: We thank the Referee and we have added/removed the wrong parts (highlighted in yellow and strikethrough and highlighted in red).
- "At 350 nm after 120 min the isomerization of t-FEF77 was complete (94.2% of c-FEF77)…" But isomerization takes 60 minutes, and then the amount does not change.
Answer: As the Referee say the isomerization takes 60 min, we have corrected the sentence.
- “To the best of our knowledge, it is the first time that a complete isomerization of a FA derivative has been observed.” It is necessary to add an explanation in what is the uniqueness of the obtained structure of FEF77, which allowed complete isomerization?
Answer: We have investigated the structural requirements for the complete isomerization of ferulic derivatives preparing about 20 compounds. The results of this study will be disclosed in a in a paper we are preparing. We can anticipate to the Referee that only tertiary amides bearing at least one aliphatic N-substituent are able to isomerized completely.
- Table 1. Why is RSA(1) = 20.3 greater than RSA(2) = 16.7 for the FA concentration of 200 M (need to be corrected by μM?)?
Answer: We thank the referee. We have corrected the value with the right number.
- Need to add the correspondence of images to the letters a), b), c) in the caption of Figure 4. Scale bars of images on Figure 4c are not visible. In Figure 4c, twice labeled 0.75 mM H2O2 + FA 12.5 µM.
Answer: Done
- Supplementary materials should be added the information on the conditions for recording UV and NMR spectra (concentrations and solvents).
Answer: Done. The concentration for UV spectra is 25
- Contributions of all authors should be added.
Answer: Contributions of all authors was uploaded during the online submission procedure. Now it is pasted also in the manuscript. Done
We have added this sentence in the manuscript (lines 432-436).
Conceptualization, Elena Grasselli and Lisa Moni; Data curation, Chiara Lambruschini, Ilaria Demori, Zeinab El Rashed, Leila Rovegno, Katia Cortese, Elena Grasselli and Lisa Moni; Investigation, Chiara Lambruschini, Ilaria Demori, Zeinab El Rashed, Leila Rovegno, Elena Canessa and Katia Cortese; Writing, original draft, Elena Grasselli and Lisa Moni; Writing, review and editing, Chiara Lambruschini and Ilaria Demori.
Reviewer 2 Report
Chiara Lambruschini et al carried out a research work on ferulic acid and feruloyl amides. The feruloyl tertiary amides were synthesized using Ugi four component reaction. The antioxidant activities of synthesized chemicals via DPPH assay. The manuscript can be published on Molecules, however the following issues should be addressed.
- For the abstract, no quantitative description on results, it would be better to provide essential data.
- For the viability assay, why the authors choose the initial concentration of 3.125 uM?
- For the cell assay, no positive control was used.
- Line 159, for EC50, the word better is not appropriate, generally for the comparison of EC50, high or low can be used.
- Line 237-246, the authors carried a thorough discussion on NAFLD, however, no relevant experiment were performed in this manuscript. How to provide the relevance of NAFLD to the synthesized chemicals? It would be revised this section.
- Please double check the manuscript for potential grammar errors and typos.
Author Response
Genova, December 23rd 2020
Dear Editor,
please find enclosed the revised version of the manuscript “Synthesis, photoisomerization, antioxidant activity and lipid-lowering effect of ferulic acid and feruloyl amides” emended according to reviewers’ suggestions. We thank the reviewers for their constructive criticism. A point by point answer to their comments follows.
Referee 2
For the abstract, no quantitative description on results, it would be better to provide essential data.
Answer: The abstract has been re-organized according to the referee’s suggestion.
For the viability assay, why the authors choose the initial concentration of 3.125 uM?
Answer: We started to consider the concentration of 50 µM as the highest to use, since we found that, at least for some cell lines (e.g. HepG2 cells) a decrease in cell viability has been reported with similar FA concentrations (Zubair H, Khan HY, Sohail A, Azim S, Ullah MF, Ahmad A, Sarkar FH, Hadi SM. Redox cycling of endogenous copper by thymoquinone leads to ROS-mediated DNA breakage and consequent cell death: putative anticancer mechanism of antioxidants. Cell Death Dis. 2013 Jun 6;4(6):e660. doi: 10.1038/cddis.2013.172.). Starting from 50 μM FA, we halved the doses until reaching 3.125 µM. As shown in the results, in our hands none of the concentrations used significantly affected the viability of HECV and FaO cells. For the sake of clarity, this is now quoted in the Results section (Paragraph 2.4.2).
For the cell assay, no positive control was used.
Answer: It is quite a common practice not to show a positive control for cytotoxicity in MTT assays (Zubair et al., Redox cycling of endogenous copper by thymoquinone leads to ROS-mediated DNA breakage and consequent cell death: putative anticancer mechanism of antioxidants. Cell Death Dis. 2013 Jun 6;4(6):e660. doi: 10.1038/cddis.2013.172; Grasselli et al., Direct effects of Bisphenol A on lipid homeostasis in rat hepatoma cells. Chemosphere 2013. 91: 1123–1129. Doi: 10.1016/j.chemosphere.2013.01.016). However, in the present paper, the treatment of HECV cells with hydrogen peroxide shown in figure 4B is indeed a positive control for cytotoxicity. The same experiment was also performed for FaO cells, even if data are not shown.
Line 159, for EC50, the word better is not appropriate, generally for the comparison of EC50, high or low can be used.
Answer: Done.
Line 237-246, the authors carried a thorough discussion on NAFLD, however, no relevant experiment were performed in this manuscript. How to provide the relevance of NAFLD to the synthesized chemicals? It would be revised this section.
Answer: As some of us reviewed (Grasselli et al., Models of non-Alcoholic Fatty Liver Disease and Potential Translational Value: the Effects of 3,5-L-diiodothyronine. Annals of Hepatology 2017, 16, 707-719, doi:10.5604/01.3001.0010.2713), the in vitro models consisting of hepatic cell lines treated with oleate/palmitate to induce lipid accumulation within hepatocytes are widely accepted NAFLD models, and they are considered valuable tools to screen a large number of compounds with putative beneficial effects against NAFLD. By using these models, our group contributed several papers demonstrating the positive effects of iodothyronine in the regulation of lipid metabolism (Demori, Voci and Grasselli. Iodothyronines as lipid-lowering agents. In: The molecular Nutrition of fats, 1st edition. Elsevier, 2018. Doi: 10.1016/B978-0-12-811297-7.00028-7).
In the present paper, we show the antioxidant properties of FEF77. This is of interest for liver steatosis, since oxidative stress is one of the multiple hits that can trigger and/or worsen NAFLD pathogenesis. Phenolic compounds often exhibit both antioxidant and antisteatotic actions that could be interpreted as two manifestations of an overlapping effect on energy metabolism (as an example, see our review on the effects of Mediterranean spices and aromas against NAFLD: Baselga-Escudero et al. Beneficial effects of the Mediterranean spices and aromas on non-alcoholic fatty liver disease. Trends Food Sci. Technol. 2017, 61, 141-159, doi:10.1016/j.tifs.2016.11.019). In accordance with this view, in the present paper we demonstrate a direct lipid-lowering effect of FEF77 in an in vitro model of NAFLD, where the mechanisms can be studied directly at the cellular level, keeping the translational value of the observed results, without exposing humans to unnecessary risks. Undoubtedly, information in this field need to be confirmed by clinical studies.
According to the referee’s suggestion, the discussion on NAFLD has been re-organized in order to highlight the significance of our results for future research on NAFLD treatment, which is a must, since no specific therapy for NAFLD has been approved so far.
Please double check the manuscript for potential grammar errors and typos.
Answer: Done
Round 2
Reviewer 1 Report
The authors made all the necessary changes.